# Impact of Barrel Toasting on Ellagitannin Composition of Aged Cognac Eaux-de-Vie

**DOI:** 10.3390/molecules27082531

**Published:** 2022-04-14

**Authors:** Mathilde Gadrat, Catherine Emo, Joël Lavergne, Pierre-Louis Teissèdre, Kléopatra Chira

**Affiliations:** 1Unité Mixte de Recherche 1366 Œnologie, INRAE, Bordeaux INP, Institut des Sciences de la Vigne et du Vin, Université de Bordeaux, CS 50008-210, Chemin de Leysotte, CEDEX, 33882 Villenave d’Ornon, France; mathilde.gadrat@u-bordeaux.fr (M.G.); pierre-louis.teissedre@u-bordeaux.fr (P.-L.T.); 2Courvoisier SAS, 2 Places du Château, 16200 Jarnac, France; catherine.emo@beamsuntory.com (C.E.); joel.lavergne@wanadoo.fr (J.L.)

**Keywords:** *C*-glucosidic ellagitannins, ellagitannin-derived spirit compounds, cognac, eau-de-vie, barrel toasting

## Abstract

It is well established that *C*-glucosidic ellagitannins contribute to wine quality, and new forms of ellagitannins have been found recently in cognac eaux-de-vie. The contribution of some ellagitannin-derived spirit compounds to eaux-de-vie taste has been demonstrated recently. However, there is a gap in our knowledge of the content, composition, and evolution of *C*-glucosidic ellagitannins in this matrix. Indeed, the quantification of these compounds and their evolutionary compounds have never before been researched in cognac eaux-de-vie. Thus, the aim of this study was not only to quantify these compounds, but also to study their kinetics and to observe how they are impacted by barrel toasting. For this purpose, barrels representing eight different toasting levels were used to age the same eau-de-vie during the first 18 months. Ellagitannin quantification was carried out by HPLC-Triple quadrupole. The results showed that the evolutionary trend of the eight ellagitannins is the same for all eight types of barrel toasting. The maximum concentrations of *C*-glucosidic ellagitannins were found after 3 months of aging (up to 23 mg/L) before decreasing to 18 months (9.7 mg/L), whereas ellagitannin-derived spirit compound concentrations increased throughout aging (up to 130.9 mg/L). In addition, barrel toasting had such an impact on ellagitannin content that barrels could be differentiated according to their levels. Eaux-de-vie in barrels with high toasting were lower in ellagitannins concentrations.

## 1. Introduction

Ellagitannins belonging to the larger hydrolyzable tannins family are the main extractible oak wood phenolic compounds, representing up to 10% of wood core dry weight [1]. *C*-glucosidic ellagitannins have a specific structure composed of an open-chain glucose esterified in position 4 and 6 by a hexahydroxydiphenoyl unit (HHDP), and a nonahydroxyterphenoyl (NHTP) unit esterified in position 2, 3, and 5. The NHTP is also linked by a *C*-glycoside bond at the carbon-1 atom to the open-chain glucose core [2]. Vescalagin (**1**) and its *C*-1 epimer castalagin (**2**) were the first *C*-glucosidic ellagitannins to be isolated and characterized, more than fifty years ago, from *Castanea sativa* (chestnut) and *Quercus sesseliflora* (oak) [3,4,5]. These monomers account for 40–60% of oak wood ellagitannins [6]. Six other ellagitannins were later isolated from the two same heartwood species: the dimers roburin A (**5**) and D (**6**), glycosylated monomers grandinin (**3**) and roburin E (**4**), and glycosylated dimers roburin B (**7**) and C (**8**) [7,8] (Figure 1).

*C*-glucosidic ellagitannins are found in various concentrations in wood. These variations depend on the result of different factors, such as botanical species, geographical origin, single-tree variability, sampling position in the tree, grain, and processing of the wood in the cooperage [9,10,11,12,13]. All these parameters will have an impact on *C*-glucosidic ellagitannin concentrations, which may also influence the sensory profile of barrel-aged wines or spirits [14,15].

Toasting is a key step in the barrel manufacturing process. Oak wood toasting leads to thermodegradation of compounds, which release new volatile and non-volatile compounds [15,16,17,18,19]. The toasting applied to the wood varies from one cooperage to another. In general, the more the toasting increases, the greater the changes in ellagitannin content and composition. More precisely, ellagitannin degradation depends strongly on the toasting temperature and time, since the above compounds are thermosensitive phenolic compounds [15,20,21].

Ellagitannins have been extensively studied in wines, especially in order to characterize their organoleptic impact [22,23,24,25,26]. The large number of free hydroxyl groups on the molecule favors interactions with salivary proteins (Figure 1). In spirits, however, total polyphenols have been studied, but very few studies have quantified individual ellagitannins [27,28,29]. Some individual ellagitannins, such as the monomers, have been quantified in different wine spirits, such as brandy [30], and a recent study has identified the eight individual ellagitannins for the first time in cognac eaux-de-vie [31]. Cognac is a prestigious French spirit, produced exclusively in Charente and Charente-Maritime and made from double-distilled white wines. Its specifications require the eau-de-vie to be aged in barrels for more than three years. However, to date, there is a lack of knowledge concerning the evolution of oak ellagitannins in cognac eaux-de-vie and their organoleptic impact.

In cognac, the oak ellagitannin monomers and dimers that have been quantified represent only a small part of the total ellagitannin concentration in spirits, suggesting that other unidentified ellagitannins may make up this total concentration [14]. In addition, previous studies on barrel-aged wines have shown that *C*-glucosidic ellagitannins were found in small amounts in these matrices, suggesting that these molecules undergo chemical transformations, via oxidation or reactions with other wine compounds, for example [32]. More particularly, β-1-*O*-ethylvescalagin (**9**) (Figure 2), which results from the nucleophilic addition of ethanol to vescalagin in acidic conditions, and β-1-*O*-ethylvescalin (**12**), which is its hydrolyzed form, have been identified and quantified in red aged wine [23,32,33]. In the case of whiskey, the oxidation of ellagitannins was responsible for the formation of new compounds called whiskey tannins B (**10**–**11**) and A (**14**–**15**), whose structures are represented in Figure 2. These compounds were isolated and identified for the first time in Japanese whiskey [34], and recently two other isomers of these compounds were detected and quantified for the first time in cognac eaux-de-vie [35]. Moreover, other ellagitannin derivatives named brandy tannin B (**12**) [35] and its hydrolyzed form brandy tannin A (**16**) [36] (Figure 2) have been identified and quantified in cognac eaux-de-vie for the first time. The above ellagitannin-derived spirits compounds have been quantified in the context of a vertical of cognac eaux-de-vie representing different vintages. However, their evolution in young eaux-de-vie has never been explored. Thus, it would be interesting to quantify them during the first 18 months of barrel aging in order to better understand *C*-glucosidic ellagitannins evolution in eaux-de-vie, and in particular to be able to observe after how long chemical transformations take place and which ones are predominant in cognac eaux-de-vie.

Given the limited knowledge of the concentrations, composition, and evolution of *C*-glucosidic ellagitannins in cognac eaux-de-vie and the recent characterization of new oxidative ellagitannin compounds, the objective of this study was to monitor and compare the extraction kinetics of the eight known oak *C*-glucosidic ellagitannins (**1**–**8**) and the new ellagitannin-derived spirit compounds (**9**–**16**) (Figure 2) in various eaux-de-vie aged in oak barrels subjected to different toasting levels. The evolution kinetics of the oak *C*-glucosidic ellagitannins were monitored for the first 18 months of aging. The influence of toasting level on *C*-glucosidic ellagitannin composition during aging of the same eau-de-vie in barrels was also studied.

## 2. Results and Discussion

### 2.1. Evolution of C-Glucosidic Ellagitannins and Ellagitannin-Derived Spirit Compounds in Cognac Eaux-de-Vie during Aging

A non-oaked eau-de-vie was placed in 32 barrels of 400 L comprising eight different toasting levels, i.e., four barrel replicates per level. The evolution of *C*-glucosidic ellagitannins was studied during eau-de-vie aging for 18 months. The concentration of the sum of the eight oak *C*-glucosidic ellagitannins (**1**–**8**) increased rapidly until arriving at a peak at 3 months (from 23 mg/L to 14 mg/L depending on the type of toasting), as was previously observed in red wine [23]. After reaching this maximum concentration, it then decreased slowly over time (at 18 months, the concentrations reached a range from 9.7 mg/L to 5.4 mg/L). Regardless of the type of barrel toasting, the trend in *C*-glucosidic ellagitannins in the eau-de-vie was the same (Figure 3A). Therefore, the content of each *C*-glucosidic ellagitannin was plotted for the different sampling times for one of the toasting levels, MT+N toasting (Figure 3B). Furthermore, during the first month of aging, only the *C*-glucosidic ellagitannin monomers castalagin and vescalagin were detected and quantified at low levels in the eau-de-vie (around 160 µg/L for castalagin and 20 µg/L for vescalagin). None of the other *C*-glucosidic ellagitannins were found. After 3 months of aging, when the eau-de-vie had extracted the maximum of the eight *C*-glucosidic ellagitannins, the content of each varied over time, but castalagin (**2**) (78% on average) was always the main *C*-glucosidic ellagitannin, followed by grandinin (**3**) (7% on average), vescalagin (**1**) (5% on average), and roburins A-E (**4**–**8**) (10% on average).

However, the trends for the compounds in the eau-de-vie during the first 18 months of aging were not the same for all of the *C*-glucosidic ellagitannins, as the content of each changed according to the contact time of the eau-de-vie with the wood. In particular, between 6 and 12 months of aging, the concentration of vescalagin (**1**) decreased more rapidly than its isomer castalagin (**2**), by 31% compared to 25% for castalagin (**2**). This decrease became more pronounced between 12 and 18 months of aging, as the loss of vescalagin (**1**) was around 59%, against only 8% for castalagin (**2**). This can be explained by the fact that vescalagin (**1**) is more reactive than its epimer, due to the position of the hydroxyl in the *C*-1 position, and can react more easily with other eaux-de-vie constituents, such as ethanol, and undergo transformations during aging, as previously observed in wine [23,32]. Furthermore, grandinin (**3**) did not appear to be very reactive, as the concentration of this compound did not vary significantly between 3 and 18 months, and its isomer roburin E (**4**) only decreased by 16% between 6 and 12 months, and between 12 and 18 months, in contrast to the dimers (**5**–**6**) and glycosylated dimers (**7**–**8**), which followed the same downward trend as vescalagin (**1**). It is probably for this reason that grandinin (**3**) is present in higher concentrations than vescalagin (**1**) in eaux-de-vie, whereas the opposite is observed in oak wood. It is possible that vescalagin (**1**) reacts very quickly in this matrix, and that its extraction rate is, therefore, slower than its reaction rate with other compounds.

These findings cannot be compared with previous in the literature as no previous data for eaux-de-vie exist. When a comparison is made with the wine matrix, the results presented are in agreement [11,23]. The maximum concentration of *C*-glucosidic ellagitannins was also reached at 3 months in barrel-aged red wines. Furthermore, levels of vescalagin (**1**) also varied more than those of castalagin (**2**), although in red wines the amount of vescalagin (**1**) was higher than that of grandinin (**3**).

Under the same conditions, formation of the ellagitannin-derived spirit compounds (**9**–**16**) from vescalagin (**1**) and castalagin (**2**) was then monitored. The sum of the concentrations of the eight ellagitannin-derived spirit compounds (**9**–**16**) during 18 months of aging is represented in Figure 4A. These derivatives can be detected in eaux-de-vie after one month of aging in oak barrels.

Regardless of toasting level, their concentrations showed the same behavior, i.e., their content increased continuously during the 18 months of aging, with the exception of LTN. Ellagitannin-derived spirit compounds levels ranged from 12.7 mg/L to 27.3 mg/L at 1 month of aging, depending always on toasting type, and finally reached concentrations ranging from 78.3 mg/L to 130.9 mg/L after 18 months. As previously achieved, the content of each ellagitannin-derived spirit compound (**9**–**16**) was plotted for the different sampling times for the eau-de-vie aged in the MT+N toasted barrel (Figure 4B). During the first 3 months of aging, the main ellagitannin-derived spirit compounds observed in the eau-de-vie were the non-HHDP-hydrolyzed ones, i.e., whiskey tannins B1 (**10**) and B2 (**11**), brandy tannin B (**12**) and β-1-*O*-ethylvescalagin (**9**), as their concentrations increased strongly (between 69% and 87%) in the first 3 months, while those without HHDP moieties, such as whiskey tannins A1 (**14**) and A2 (**15**), brandy tannin A (**16**) and β-1-*O*-ethylvescalin (**13**), increased by between 37 and 49%. On the contrary, during the second phase of these kinetics, and more precisely between 9 and 18 months, the concentrations of whiskey tannins A1 (**14**) and A2 (**15**) and those of brandy tannin A (**16**) increased more rapidly (between 36 and 40%) than those of whiskey tannins B1 (**10**) and B2 (**11**), and those of brandy tannin B (**12**) (by between 23 and 28%). In addition, β-1-*O*-ethylvescalagin (**9**) and β-1-*O*-ethylvescalin (**13**) concentrations increased more slowly, by around 11% and 13%, respectively. However, although the total concentration of the ellagitannin-derived spirit compounds increased according to their aging time, their individual proportions followed the same trend. In particular, whiskey tannins B2 (**11**) and A2 (**15**) were the main ellagitannin-derived spirit compounds among those studied (35% and 20% on average, respectively), while the other compounds represented 44% on average (accounting for 6 to 9% each).

As no studies have focused on *C*-glucosidic ellagitannin in spirits, the levels found here for the eight known *C*-glucosidic ellagitannins (**1**–**8**) and the new ellagitannin-derived spirit compounds (**9**–**16**) constitute the first observations. Previous works investigating trends in vescalagin derivatives concerned only red wines [23,31]. β-1-*O*-ethylvescalagin (**9**) along with flavano-ellagitannins named acutissimins and epiacutissimins, were identified and quantified. β-1-*O*-ethylvescalagin (**9**) concentration increased during the first year of aging, following the same behavior as in the wine [23,31], whereas its concentration in a Bordeaux red wine after 18 months of aging is around 0.85 mg/L [32]. The levels of this compound ranged from 4.30 to 8.96 mg/L in the eaux-de-vie and were higher than those found in the wine for the same contact time with the oak wood and were dependent on barrel toasting level. Since the eau-de-vie has a higher alcohol content (63% *v*/*v*) than wine, the reaction between vescalagin (**1**) and ethanol may occur more quickly and easily in this matrix, even though the pH of the spirits is higher than that observed in red wine. Moreover, in a recent work with multiple commercial cognac eaux-de-vie and multiple vintages ranging from 1986 to 2020 [35], β-1-*O*-ethylvescalagin (**9**) was not found in commercial eaux-de-vie and its mean concentration for the vintages between 1986 to 2020 was around 1 mg/L, which is still higher than in wine [23,32,37]. The differences found between the previous study [35] and this one may be due to oak wood variability. The initial oak wood vescalagin (**1**) level is influenced notably by the type and length of both seasoning and toasting steps during the barrel manufacturing process, as well as by the eau-de-vie aging process used in the cellars. In practice, the eaux-de-vie are often transferred to used barrels after a few months of aging, resulting in lower ellagitannin extraction by the eaux-de-vie, depending on the duration of use of the barrels.

Concerning whiskey tannins, the levels of their isomers B1 (**10**) and B2 (**11**) found in this study were always higher than their corresponding hydrolyzed isomers A1 (**14**) and A2 (**15**). However, in a previous eaux-de-vie study with different vintages, whiskey tannins B1 (**10**) and B2 (**11**) were in smaller quantities than their hydrolyzed forms [35]. Although there is hydrolysis during aging and whiskey tannins B1 (**10**) and B2 (**11**) can be hydrolyzed to form whiskey tannins A1 (**14**) and A2 (**15**), respectively, the kinetics of this reaction may be slower than that of whiskey tannins B1 (**10**) and B2 (**11**) formation from the oak ellagitannin, which may explain their higher values found in this work. Moreover, during long aging, there is no more castalagin (**2**) or vescalagin (**1**) to be extracted from the wood, and, therefore, the B1 (**10**) and B2 (**11**) whiskey tannins can no longer be formed. Two other compounds resulting from the transformation of vescalagin (**1**) have been identified and characterized recently in cognac eaux-de-vie [35,38]. Brandy tannins A (**16**) and B (**12**) are formed from vescalagin derivative β-1-*O*-ethylvescalagin (**9**) after an oxidation reaction followed by a second addition of ethanol. Brandy tannin A (**16**) is present in concentrations slightly lower than brandy tannin B (**12**), as it can be formed by brandy tannin B hydrolysis (**12**) and also by β-1-*O*-ethylvescalin (**13**) oxidation following ethanol addition. The concentrations of both brandy tannins **12** and **16** were always higher than those of β-1-*O*-ethylvescalagin (**9**) and β-1-*O*-ethylvescalin (**13**). This could be explained by the fact that evolutionary reactions occur rapidly in an alcohol-rich environment. In addition, brandy tannins **12** and **16** are present at concentrations around 10 mg/L after 18 months of aging. These results suggest that they may influence cognac organoleptic quality, since it has previously been observed that these compounds could have an impact on eaux-de-vie taste at 2 mg/L for brandy tannin A (**16**) and at 5 mg/L for brandy tannin B (**12**) [35,38].

### 2.2. Impact of the Barrel Toasting Level on Concentrations of the C-Glucosidic Ellagitannins and Ellagitannin-Derived Spirit Compounds during Oak Wood Aging

The impact of barrel toasting level on *C*-glucosidic ellagitannin content and composition was also investigated. As eight different toasting levels were used, the *C*-glucosidic ellagitannin concentrations were represented for each toasting level at 3 months of aging, corresponding to the time when eau-de-vie extraction reaches its maximum amount of *C*-glucosidic ellagitannins, as mentioned above (Figure 5). As the castalagin levels were much higher than those of the other ellagitannins, a zoom on a more suitable scale was shown for the other seven compounds, in order to better observe the significant differences related to the type of toasting of the barrel. The toasting levels studied represented four different toasting temperatures (light, medium, medium +, and high) with two different durations (slow and normal). The ANOVA of the temperature-duration interaction was performed for each compound studied. If the *p*-value of the interaction was *p* ≤ 0.05, then both factors had to be treated together and, therefore, toasting was considered a single parameter. As the interaction was always significant, the impact of all eight toasting levels was studied for each compound. The results showed that all eight *C*-glucosidic ellagitannins were impacted by the toasting level. In general, high toasting levels (HTN and HTS) resulted in lower ellagitannin contents, as the oak wood ellagitannins are sensible to high temperatures [15,39]. However, there was no clear differentiation between the eight toasting levels at 3 months of aging, except for castalagin where concentrations were in the majority. The confidence intervals, represented on the histograms corresponding to a percentage error of 5%, were very high, demonstrating the need for biological replicates, as the inter-individual variability of the wood is substantial and may, in some cases, override the factors being studied.

Similarly, the impact of toasting level on ellagitannin-derived spirit compounds concentrations was studied at 18 months of aging, when their concentrations were the highest. The individual concentrations of the eight ellagitannin-derived spirit compounds were plotted for each toasting level at this contact time (Figure 6). These results evidenced that the barrel toasting level also had an impact on these compounds’ content, as they were formed from the oak *C*-glucosidic ellagitannins, which are sensitive to heat. The toasting level, therefore, influenced their concentrations indirectly by having previously caused differences in the extraction of ellagitannins from the wood.

Longer aging seems to have accentuated the differences in *C*-glucosidic ellagitannin concentrations according to toasting level, because although the *C*-glucosidic ellagitannins were transformed over time, levels of ellagitannin-derived spirit compounds during aging were strongly dependent on toasting level. The differences observed among the toasting levels were more pronounced after 18 months, and it seems that toasting length has a strong impact on these compounds’ levels. Furthermore, a high toasting temperature favored β-1-*O*-ethylvescalin formation (**13**), as this was the only ellagitannin derivative whose content increased significantly with toasting.

For a more overall view of toasting level impact on *C*-glucosidic ellagitannin behavior, a principal component analysis (PCA) was carried out at 18-month contact time, taking into account all of the ellagitannins found to date in cognac eaux-de-vie. The distribution map of the individuals according to their toasting level was plotted (Figure 7A), with the correlation circle of the associated variables (Figure 7B). More precisely, the PCA was on two dimensions, explaining 65.3% of the variance. Dimension 1 (35.5%) was explained by the eight oak *C*-glucosidic ellagitannins (**1**–**8**) and dimension 2 (29.8%) was represented by the ellagitannin-derived spirit compounds (**9**–**16**).

Barrel toasting level had a significant impact on ellagitannin levels, as clear differences were observed between toasting levels. As observed previously, PCA clearly showed that β-1-*O*-ethylvescalin (**13**) levels were higher for the HTN high toast, in contrast to the other compounds. The MTN and MTS medium toasts were the least distinguishable from the others in terms of ellagitannin content extracted by eau-de-vie. However, the LTS and MTS samples were the richest in whiskey tannins and in brandy tannins.

## 3. Materials and Methods

### 3.1. Chemicals

Ultrapure water (Milli-Q purification system, Millipore, France) was used. Methanol, water, and formic acid used for chromatographic separation were LC-MS grade and purchased from Agilent Chemical (France). Oak *C*-glucosidic ellagitannins, such as vescalagin (**1**), castalagin (**2**), grandinin (**3**), and roburin A-E (**4**–**8**) were extracted from *Q. robur* heartwood and purified (>95% pure), as previously described [40]. These standards were kindly provided by Pr. S. Quideau. Whiskey tannin B1 (**10**), B2 (**11**) and brandy tannin B (**12**) were obtained as described previously [35].

### 3.2. Cognac Eaux-de-Vie and Sample Preparation

Thirty-two oak wood barrels of 400 L were manufactured with different toasting levels. Eight different toasting levels were studied, and four barrel replicates were produced and used for each level. A young, unwooded eau-de-vie from double-distilled *Grande Champagne terroir* with an alcohol content of 63% (*v*/*v*) was placed in these 32 barrels. Changes in the eau-de-vie were monitored during the first 18 months of aging. The eight toasting levels studied consisted of four toasting temperatures (light, medium, medium plus, and high) and two toasting durations (normal and slow) for each temperature. The description of the different toasting levels is in Table 1. The values presented were obtained from the data recorded during the toasting of the barrel. Indeed, the temperatures were recorded throughout the toasting process and the values given (Table 1) represent the measured sum of temperature (°C) based on the time that the entire toasting period lasts. Eau-de-vie were sampled at 1 month (T1), 3 months (T2), 6 months (T3), 9 months (T4), 12 months (T5), and 18 months (T6) of aging.

The procedure for the preparation of eaux-de-vie samples prior to HPLC-UV-MS analysis was adapted from the procedure described previously [31]. Each eau-de-vie sample (100 mL) was evaporated under vacuum, and the dried residue was dissolved in 500 µL of milli-Q water with 0.1% of formic acid and filtered through 0.45 µm filters prior to HPLC-UV-MS analysis, in order to quantify the oak *C*-glucosidic ellagitannins in cognac eaux-de-vie. A second method for the preparation of eaux-de-vie samples was used for ellagitannin-derived spirit compound quantification. As these compounds are less soluble in water, the eaux-de-vie samples were injected directly into the HPLC-UV-MS after being diluted with milli-Q water by a factor of 5 and filtered through 0.45 µm filters.

### 3.3. HPLC-UV-QQQ Analysis

For the quantification of *C*-glucosidic ellagitannins in cognac eaux-de-vie, the method had been developed and validated beforehand, as described previously [31]. The HPLC-UV system was an Agilent 1200 Infinity series (Agilent Technologies, Waldbronn, Germany). For ellagitannin quantification, a Kinetex column (150 × 3 mm, 2.6 μm particle size, Phenomenex, Le Pecq Cedex, France) was used with acidified water at 0.1% formic acid (Eluent A) and acidified methanol at 0.1% formic acid (Eluent B) as mobile phases. The gradient elution of solvent B was as follows: 3% from 0 to 2 min; 3 to 3.5% from 2 to 6 min; 3.5% from 6 to 7.5 min; 3.5 to 4% from 7.5 to 9 min; 4 to 4.5% from 9 to 11 min; 4.5 to 5% from 11 to 13 min; 5 to 10% from 13 to 16 min; 10 to 15% from 16 to 19 min; 15 to 20% from 19 to 22 min; 20 to 30% from 22 to 38 min; 30 to 40% from 28 to 33 min; 40 to 99% from 33 to 35 min; 99% from 35 to 39 min; and then the HPLC column was equilibrated for 4 min using the initial conditions before the next injection. A flow rate of 400 µL/min was applied with a detection at 280 nm. The HPLC-UV system was coupled with a 6460 triple-quadrupole mass spectrometer (QQQ) equipped with a heated electrospray ionization probe (both from Agilent Technologies, Waldbronn, Germany). Electrospray ionization and mass detection were performed in negative mode with the following parameters: gas temperature and flow were 350 °C and 5 L/min, respectively; sheath gas temperature and flow were 250 °C and 10 L/min, respectively; capillary voltage was 4500 V. For ellagitannins quantification, acquisition was performed in Single Ion Monitoring (SIM) mode, with a fragmentor value at 135 V and a cell accelerator voltage at 8 V. MassHunter qualitative analysis software was used for the acquisition process, and MassHunter quantitative analysis was used for data treatment (both from Agilent Technologies, Waldbronn, Germany).

### 3.4. Calibration Curves

Quantification of the eight *C*-glucosidic ellagitannins (**1**–**8**) was obtained from standard calibration curves. The calibration curves were obtained by injecting each *C*-glucosidic ellagitannin (**1**–**8**) in increasing concentrations ranging from 0.1 mg/L to 20 mg/L. Therefore, each of the eight *C*-glucosidic ellagitannins was quantified according to its own calibration curve. Stock solutions of β-1-*O*-ethylvescalagin (**9**), brandy tannin B (**12**) and whiskey tannin B1 (**10**) and B2 (**11**) (two isomers forms) (1 g/L for each compound) were also prepared. The calibration curves were performed from 0.1 mg/L to 20 mg/L. The concentrations of β-1-*O*-ethylvescalin (**13**) were expressed as β-1-*O*-ethylvescalagin equivalents. In the same way, brandy tannin A (**16**), whiskey tannin A1 (**14**) and A2 (**15**) were expressed as brandy tannin B (**12**), whiskey tannin B1 (**10**) and B2 (**11**) equivalent, respectively. The quantification of each compound was performed using their molecular ion, as shown in Table 2.

### 3.5. Statistical Treatment

All of the statistical treatments were performed using Rstudio software (Rstudio Inc., Boston, MA, USA, 2018). Normality and homocedasticity of the residuals were evaluated for all parameters, using the Shapiro–Wilk test and Levene’s test, respectively. As populations were normally distributed and presented homogeneity in variance, parametric tests were used. The data was, therefore, submitted to two-way ANOVA (toasting temperature x toasting duration). If the interaction *p*-value was statistically significant (*p* ≤ 0.05), a Tukey test was directly applied to evaluate the degree of the significant differences. If not, prior to the Tukey evaluation, a one-way ANOVA was run individually for data corresponding to each significant factor. If parametric tests could not be used, non-parametric tests were performed using Kruskal–Wallis and pairwise Wilcoxon as post hoc. Principal Component Analysis (PCA) was applied as a multivariate statistical tool to examine any possible grouping of samples according to toasting.

## 4. Conclusions

For the first time, the evolution of sixteen *C*-glucosidic ellagitannins was followed in cognac eaux-de-vie during 18 months of aging. The influence of the barrels toasting level on the ellagitannin content and their behavior was also studied.

The eight oak *C*-glucosidic ellagitannins (**1**–**8**) concentrations increased during the first three months and then decreased gradually afterwards regardless of the type of toasting. Regarding the attitude of evolutionary *C*-glucosidic ellagitannins, it is different depending on compound structure. Ellagitannin-derived spirit compounds (**9**–**16**) levels increased with aging time. During the first 3 months of aging, their non-hydrolyzed forms, such as whiskey tannin B, brandy tannin B and β-1-*O*-ethylvescalagin (**9**), were predominant. After 9 months, however, their hydrolyzed forms, such as whiskey tannins A1 (**14**), A2 (**15**) and brandy tannin A (**16**), formed faster. Concerning toasting influence, toasting length had a strong impact on these compounds, mainly after 18 months. Slow toasting gave rise to more ellagitannin extraction than normal toasting. Overall, a clear toasting differentiation was observed according to ellagitannin content for light and high toasting samples, as the eau-de-vie in the light toasted barrel presented two times more ellagitannins than the eau-de-vie in the high toasted barrel. On the other hand, HTN samples indicating the highest β-1-*O*-ethylvescalin (**13**) levels; suggesting that this form of ellagitannins is more resistant to high temperatures. This work gives the first insights for the impact of barrel toasting on the evolution of ellagitannins in cognac eaux-de-vie. It would be interesting to follow the kinetics for a further three years, which corresponds to the minimum cognac aging period. The goal would be to see whether some ellagitannin-derived spirit compounds, such as brandy tannins A and B, which contribute to Cognac organoleptic quality, could be stable over time. The type of toasting could eventually be chosen according to the type of ellagitannin-derived spirit compounds desired, notably according to their organoleptic impact and the desired quality of the cognac.

## Figures and Tables

**Figure 1 molecules-27-02531-f001:**
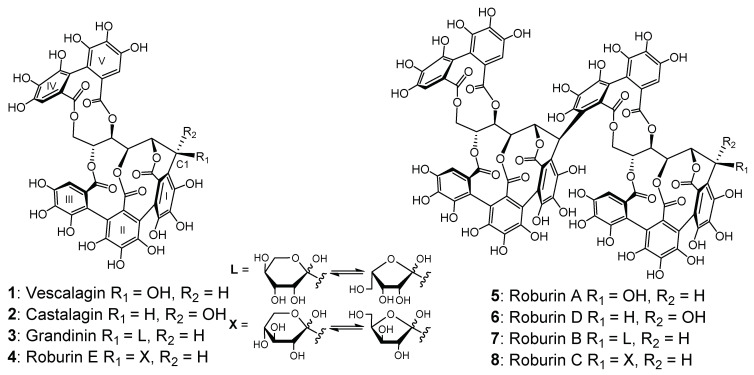
Structure of the eight *C*-glucosidic ellagitannins.

**Figure 2 molecules-27-02531-f002:**
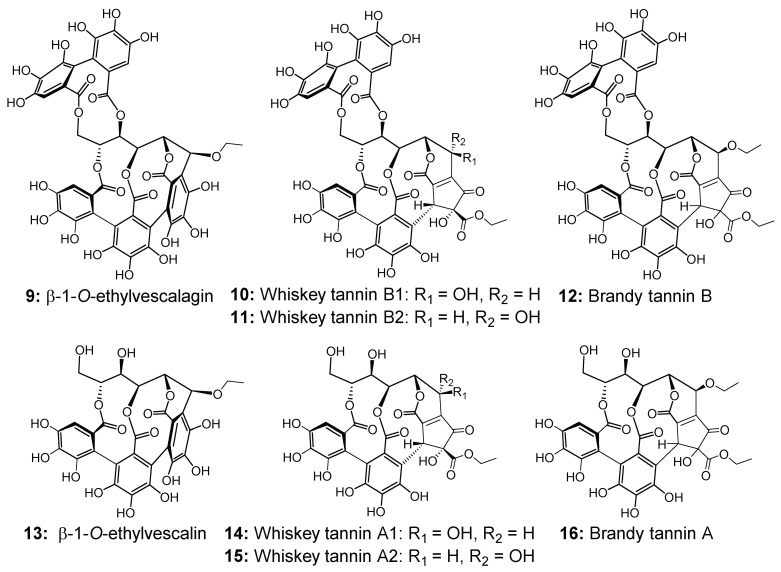
Structure of ellagitannin-derived spirit compounds (**9**–**12**) and their hydrolysis form (**13**–**16**) identified and quantified in cognac eau-de-vie aged in contact with oak wood.

**Figure 3 molecules-27-02531-f003:**
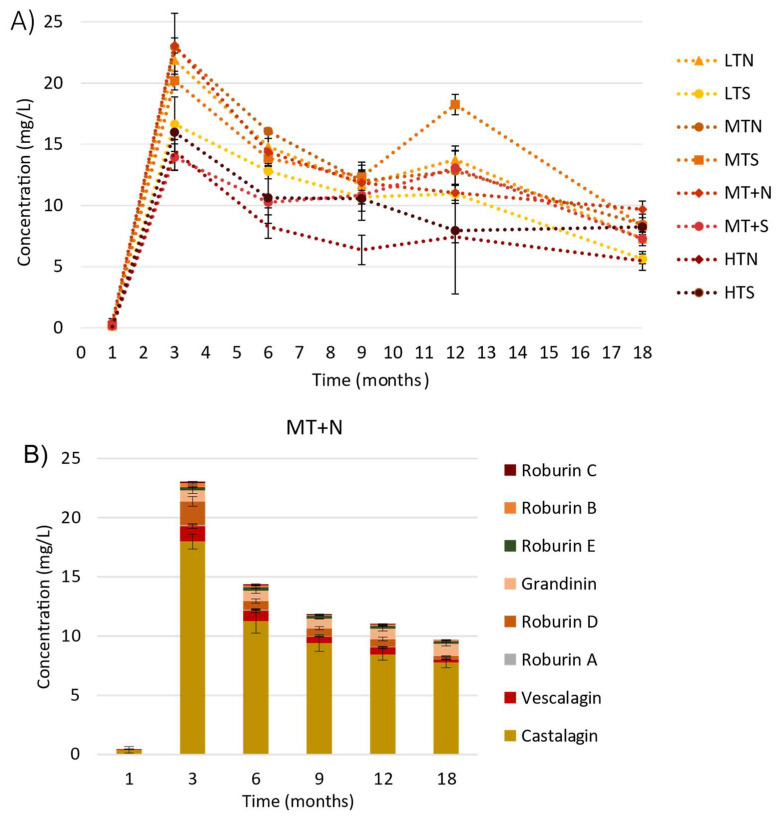
Sum of the *C*-glucosidic ellagitannin (**1**–**8**) concentrations in eau-de-vie during 18 months of aging in oak barrels manufactured with different toasting levels (LT: Light Toasting; MT: Medium Toasting; MT+: Medium Toasting Plus; HT: High Toasting; N: Normal; S: Slow) (**A**), and *C*-glucosidic ellagitannins (**1**–**8**) behavior for MT+N toasting (**B**).

**Figure 4 molecules-27-02531-f004:**
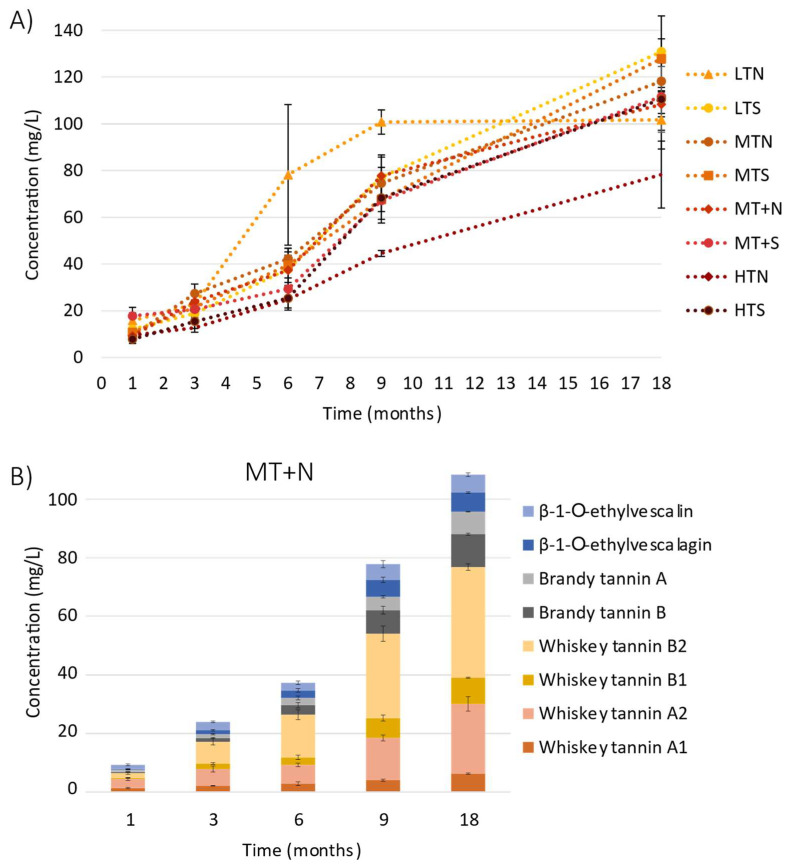
Sum of ellagitannin-derived spirit compound (**9**–**14**) concentrations in eau-de-vie during 18 months of aging in oak barrels manufactured with different toasting levels (LT: Light Toasting; MT: Medium Toasting; MT+: Medium Toasting Plus; HT: High Toasting; N: Normal; S: Slow) (**A**) and behavior of these compounds in the eau-de-vie for MT+N toasted barrel (**B**).

**Figure 5 molecules-27-02531-f005:**
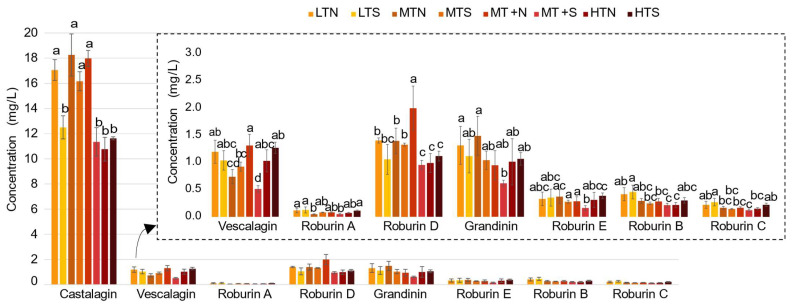
Impact of barrel toasting (LT: Light Toasting; MT: Medium Toasting; MT+: Medium Toasting Plus; HT: High Toasting; N: Normal; S: Slow) on concentrations of the *C*-glucosidic ellagitannins (**1**–**8**) in eau-de-vie at 3 months of aging. Error bar represents the confidence interval with a threshold of 0.05. Lower case letters a–d show significant differences among the different toasting modalities for each *C*-glucosidic ellagitannin (*p*
*<* 0.05).

**Figure 6 molecules-27-02531-f006:**
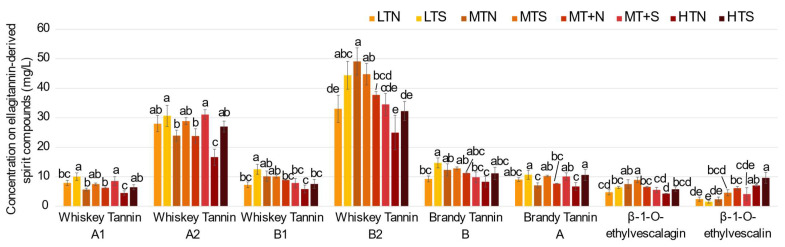
Impact of barrel toasting (LT: Light Toasting; MT: Medium Toasting; MT+: Medium Toasting Plus; HT: High Toasting; N: Normal; S: Slow) on concentrations of the ellagitannin-derived spirit compounds (**9**–**16**) in eau-de-vie at 18 months of aging. Error bar represents the confidence interval with a threshold of 0.05. Lower case letters a–e show significant differences among the different toasting modalities for each ellagitannin-derived spirit compound (*p*
*<* 0.05).

**Figure 7 molecules-27-02531-f007:**
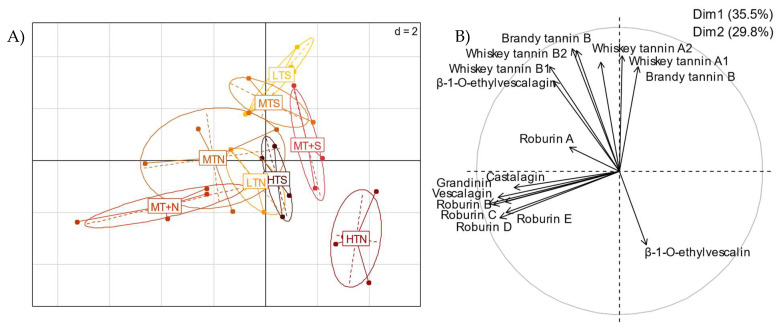
Principal component analysis (PCA) on two dimensions after 18 months of aging representing (**A**) the distribution map of the individuals according to toasting level (LT: Light Toasting; MT: Medium Toasting; MT+: Medium Toasting Plus; HT: High Toasting; N: Normal; S: Slow) and (**B**) the correlation circle of the associated variables.

**Table 1 molecules-27-02531-t001:** Description of the types of barrel toasting levels.

Toast	Notation	Sum °C.min *
Light toast normal	LTN	3560
Light toast slow	LTS	5036
Medium toast normal	MTN	4566
Medium toast slow	MTS	5134
Medium toast plus normal	MT + N	5088
Medium toast plus slow	MT + S	6015
High toast normal	HTN	6892
High toast slow	HTS	7320

* Sum of the temperatures measured as a function of time over the entire toasting period.

**Table 2 molecules-27-02531-t002:** HPLC retention times and mass fragmentation patterns of the reference compounds (**1**–**16**).

Compounds	Retention Time (min)	*m*/*z*
Vescalagin (**1**)	8.0	933 ^a^, 915, 613, 301
Castalagin (**2**)	13.5	933 ^a^, 915, 613, 301
Grandinin (**3**)	5.6	1065 ^a^, 915, 613, 301
Roburin E (**4**)	7.1	1065 ^a^, 915, 613, 301
Roburin A (**5**)	3.9	1849, 933, 924 ^a^, 915, 301
Roburin D (**6**)	5.8	1849, 933, 924 ^a^, 915, 301
Roburin B (**7**)	3.9	1981, 1065, 990 ^a^, 915, 301
Roburin C (**8**)	4.2	1981, 1065, 990 ^a^, 915, 301
β-1-*O*-ethylvescalagin (**9**)	21.2	961 ^a^, 915, 480, 301
Whiskey tannin B1 (**10**)	24.0 and 26.4	977 ^a^, 933, 675, 631, 301
Whiskey tannin B2 (**11**)	24.0 and 26.4	977 ^a^, 933, 675, 631, 301
Brandy tannin B (**12**)	32.5	1005 ^a^, 703, 301
β-1-*O*-ethylvescalin (**13**)	21.4	659 ^a^
Whiskey tannin A1 (**14**)	4.9 and 8.6	675 ^a^
Whiskey tannin A2 (**15**)	4.9 and 8.6	675 ^a^
Brandy tannin A (**16**)	27.7	703 ^a^

^a^ Ion used for the quantification.

## Data Availability

There are no publicly archived datasets analyzed or generated during the study.

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
