# Peer review of "Impact of Barrel Toasting on Ellagitannin Composition of Aged Cognac Eaux-de-Vie"

_molecules, 2022, doi:10.3390/molecules27082531_

Round 1
Reviewer 1 Report
In this study, the authors have investigated the impact of barrel toasting on ellagitannin composition of aged 2 Cognac eaux-de-vie. Research has a low novelty; scientific explanations should be made on this subject.
The abstract should be revised. In the abstract, striking sentences emphasizing the work should be added. In addition, the numerical data obtained in the study should be mentioned.
Introduction: This section contains the review of some researches, but the researches are not enough to explain the research gap clearly which results in the lack of study’s novelty. Research gaps should be more clearly highlighted.
Structure of ellagitannins is given informative, the chemical/sensory point of view should be considered.
Material and methods: The characterization of Cognac in terms of phenolic compounds should be given.
Is it new HPLC-UV-QQQ method developed and validated for ellagitannin quantification? If the method is new, all the performance of the method should be shown (not calibration curve only).
Result and discussion: Scientific explanations should be extensively made on this subject.
Conclusion is missing some relevant results obtained from the current study.
Author Response
The manuscript has been revised according to your suggestions
please see the attachement
Thank you

Reviewer 2 Report
Authors describe toasting impact on the ellagitannin composition of aged Cognac eaux-de-vie. By the way, it is very difficult to follow the results. There are many problems in figures, such as 2~6. I can not read the Fig. 2, because the displays in the MS are very confusing. Also, other figures are not self-sufficient. About toasting conditions, do authors provide enough information to readers? Thank you.
Author Response
Thank tou for the suggestions
The manuscript has been changed accordingly
please see the attachment

Reviewer 3 Report
Molecules
Impact of barrel toasting on ellagitannin composition of aged 2 Cognac eaux-de-vie.
Line 14 the aim of this study
Line 15 delete “to study”
Line 18 Indicate the abbreviation
Lines 18-20, mention the content of C-glucosidic ellagitannins in the HPLC profile and track the kinetics of ellagitannins during the eight toasting levels
Add clear recommendations at the end of the abstract
Rewrite keywords
Your manuscript doesn’t follow the exact format style. i.e., introduction move it directly in the empty space after keywords
Line 46, these variations depend on
Begin “Toasting is a key step…. As a new paragraph
Lines 67-74, bad format, adjust
I suggest moving figures 1,2 to supplementary materials as S1-S2, only mention the names of the compounds, then reorder the figures all manuscript
The materials and methods section was well described
Line 299 Thirty-two oak wood barrels of 400 L
Lines 112-113, delete the brackets around values, check all manuscript
Separate the numerical value of the unit as 7 %, check all manuscript
The discussion has a lot of potential for improvement since the results are interesting and can be very impactful
Figure 5 divided into two subfigures A (Robrin A,B, C, D, E), B (other ellagitannins)
Also, figure 6 need to be divided into 3 subfigures (whisky), (brandy), (compounds)
Enhance conclusion and don’t mention results
Some linguistic errors need to be checked all manuscript
All journal names should be italic
Update references
Carefully check the outputs of all references
Author Response
Thank you for your suggestions
The manuscript has been modified accordingly
please find here the attachment

Round 2
Reviewer 1 Report
The authors gave satisfactory answers to the points I raised.
Reviewer 3 Report
No more correction, Now, it can be accepted